# The Thoracic Inlet Heart Size, a New Approach to Radiographic Cardiac Measurement

**DOI:** 10.3390/ani13030389

**Published:** 2023-01-24

**Authors:** David Marbella Fernández, Verónica García, Alexis José Santana, José Alberto Montoya-Alonso

**Affiliations:** 1Faculty of Veterinary Medicine, University of Las Palmas de Gran Canaria, 35413 Las Palmas de Gran Canaria, Spain; 2Anicura Albea Small Animal Hospital, 35010 Las Palmas de Gran Canaria, Spain; 3Internal Medicine, Faculty of Veterinary Medicine, Research Institute of Biomedical and Health Sciences (IUIBS), University of Las Palmas de Gran Canaria, 35413 Las Palmas de Gran Canaria, Spain

**Keywords:** thoracic inlet, right lateral radiographic projection, cardiac size, dog breed, interobserver variability

## Abstract

**Simple Summary:**

The present study investigates the hypothesis that the thoracic inlet is a suitable reference point for the assessment of the heart size on a right lateral radiographic projection in dogs. This is important, since the most known method of measuring the heart size on dogs’ thoracic radiographs, the vertebral heart size, has shown breed variation and is affected by vertebral malformations as well as measurements’ transformation into vertebral units. The shortest thoracic inlet length measured from the craniodorsal manubrium to the cranioventral first thoracic vertebrae was considered. The long and short heart axes lengths were measured as described by Buchanan and Bücheler in 1995, and their total was divided by the thoracic inlet length. We found this method to be a feasible, reliable, and reproducible way of measuring a dog’s heart size on thoracic radiographs.

**Abstract:**

In 1995, the Vertebral Heart Size (VHS) method for measuring the cardiac silhouette on thoracic radiographs was published, becoming a quantifiable and objective reference way of assessing the heart size. Since then, many studies have showed that VHS is influenced by breed variations, vertebral malformations, reference points selection, and short and long axes dimensions conversion into vertebral units. The Thoracic Inlet Heart Size (TIHS) normalizes heart size to body size using the thoracic inlet length. The lengths of the long and short axes of the heart of 144 clinically normal dogs were measured on right lateral thoracic radiographs. The sum of both measures was indexed to the thoracic inlet length. For comparison, dogs of the most represented breeds in our hospital were selected to measure their heart size using the TIHS protocol. The mean TIHS value for the population studied was 2.86 ± 0.27, and 90% of dogs had a TIHS value of less than 3.25. There was no difference in TIHS between male and female, and between small and large dogs (*p*-value < 0.01). There was no difference in the TIHS value between Yorkshire Terrier, Chihuahua, and Labrador retriever breeds, and between each of those three breeds and the general population. The TIHS is a simple, straightforward and accurate way to measure heart size.

## 1. Introduction

Echocardiography is the gold standard for the evaluation of the cardiac structures when a cardiac disease is suspected based on a clinical exam finding (e.g., a heart murmur, cyanosis), clinical history information (exercise intolerance, syncope), or an abnormal cardiac silhouette on thoracic x-rays [1].

Thoracic radiography is widely available [2] and is reliable for the evaluation of generalized cardiomegaly and chamber enlargement as left atrial enlargement [3]. Different radiographic methods to determine the cardiac size have been reported: intercostal spaces, cardiothoracic ratios, Vertebral Heart Size (VHS), Vertebral Left Atrial Size (VLAS), Radiographic Left Atrial Dimension (RLAD), and Manubrium Heart Score (MHS). The intercostal spaces method, described more than five decades ago [4], is still used today (first author Poster presentation at 27th Federation of European Companion Animal Veterinary Associations Congress, Prague 2022) but has several limitations, including variations in the cardiac axes, thorax conformation, respiratory cycle, superimposition of ribs, and imprecise reference points [5]. Determining the cardiothoracic ratio is affected by variations in the thoracic conformation among dog breeds [6], and requires specific software, limiting its use in clinical practice [7]. 

The VHS method, first described in 1995 [6], normalized the sum of the cardiac long and short axes to the length of the midthoracic vertebrae, with a reference value of 9.7 ± 0.5 v (v vertebral units), suggesting an upper limit for the normal heart size in most breeds of ≤10.5 v. This study considered that different breeds might have different normal upper limits, and breed-specific studies would be required to determine more precise values for individual breeds [6]. Since then, many studies have been published proposing breed-specific VHS reference values [8,9,10,11,12,13,14,15,16,17,18,19,20,21,22,23,24,25,26]. Additionally, potential sources of VHS variation are thoracic vertebral anomalies, interobserver differences in reference point selection, and transformation into vertebral units [27]. Radiography can be necessary when echocardiography is not available, but the clinician must be aware of the variation in dog thoracic conformations and breed differences in normal VHS [28]. VLAS [29] and RLAD [30] are two methods for detecting left atrial enlargement. Both techniques normalize the left atrial size to the vertebral body length starting at the fourth thoracic vertebrae. A VLAS value ≥ 3 in the absence of echocardiography likely identifies dogs with Stage B2 mitral valve disease [28]. RLAD has demonstrated high sensitivity and specificity for detecting left atrial enlargement [29]. As with VHS, in addition to breed-specific differences, variability between individual observers and different levels of expertise may cause considerable differences between VLAS and RLAD measurements [25].

To eliminate some of the problems associated with the VHS and cardiothoracic ratios in some dogs, the MHS was proposed [3]. The manubrium was selected because it is prominent, regularly elongated, easily identified, and can be readily measured on lateral thoracic views. However, dogs with an abnormally shaped manubrium, or when its cranial margin could not be identified, were excluded from the study. Also, no echocardiographic examinations were performed to rule out subclinical cardiac disease, nor was interbreed variation assessed. 

The thoracic inlet length has been proposed as a reference point to assess tracheal diameter in brachycephalic and non-brachycephalic dogs [31,32,33]. Its use to normalize cardiac size could overcome some of the limitations related to the different methods described previously: vertebral malformations, conversion to vertebral units, manubrium malformations, and breed variation.

The objective of this prospective study was to establish a method to assess the heart size radiographically by measuring the cardiac long and short axes normalized by the thoracic inlet length on a thoracic right lateral radiographic view of clinically normal dogs. To determine whether sex and weight had an impact on the TIHS value, correlation of TIHS with VHS, as well as intra- and interobserver agreement were assessed. We hypothesized that TIHS is a simple, reliable, and reproducible method to assess cardiac size in a study population of healthy dogs. We also hypothesized that TIHS is not dependent on sex or body weight. We finally propose a TIHS reference value for healthy dogs.

## 2. Material and Methods

### 2.1. Animals

The study design was a prospective observational investigation. Dogs admitted to Anicura Albea Small Animal Hospital from March 2021 to September 2022 were studied. The selected population included client-owned dogs older than 1 year of age, with no history or concurrent clinical or radiographic signs of cardiovascular or respiratory diseases. Informed consent was obtained from the owners. Patient data including breed, sex, age, and body weight were recorded. Any patients with a heart murmur, or rhythm abnormalities other than respiratory sinusal arrhythmia on auscultation, were excluded. Dogs that had a positive response to a heartworm antigen test were excluded. A basic ultrasound exam was performed on every patient. A Vivid iq portable ultrasound machine (General Electric Medical Systems, Jiangsu, PR China) was used to acquire the images that were analyzed on an Echopack DICOM viewing system. Visual assessment and measurement of standard-echocardiographic parameters in two-dimensional (2D-) Mode, M-Mode, and Doppler Mode were carried out on the right parasternal long-axis four chamber and five chamber views, right parasternal short-axis view, and left apical and cranial views. From the right parasternal short-axis view, the left ventricle internal diameter at end diastole index to body weight (LVIdN), and left atrium aortic valve ratio (LA/Ao), were calculated. Dogs with cardiac disease based on echocardiography (including valvular abnormalities, cardiac chamber enlargement, LVIdN ≥ 1.7 and/or LA/Ao ≥ 1.6, or heartworm detected on the pulmonary artery or heart chambers) were excluded. 

### 2.2. Radiography

Digital radiographs were acquired using the same digital radiographic unit (Intech’s Veterinary Digital DR System Model Futura 10, La Cartuja Baja, Zaragoza, Spain) and retrieved with an image archiving communication system (IntechForView 12.5.1.1, La Cartuja Baja, Zaragoza, Spain). A right lateral projection was acquired for each dog as it is the usual projection taken at our hospital. Most dogs also had a ventrodorsal and left lateral projection acquired. kVP and mAs were selected for each dog based on a radiographic technique chart. Subjects with pulmonary, airway, and/or cardiovascular abnormalities, as well as those with a history of neck or chest surgery, were excluded from the investigation. The investigated thoracic radiographic views were assumed to be taken at the time of peak inspiration, and without sedation or anesthesia. The thoracic limbs were pulled as cranially as possible, to minimize superimposition with the cranial thorax. Dogs with thoracic vertebral malformations were not excluded from the study. Right lateral thoracic radiographs on which the cranial portion of the manubrium could not be identified, or the cardiac silhouette was not clearly defined, or the image was blurred due to motion artifact, were excluded.

The VHS was obtained as described by Buchanan et al. [5] and modified according to Jepsen-Grant et al. [14]. Briefly, the long axis of the cardiac silhouette was measured, starting from the central and ventral border of the carina to the most distant point of the cardiac apex. The short axis was drawn at a 90° angle to the long axis and at the level of the ventral intersection of the caudal vena cava and the cardiac silhouette. The measurements of the two axes were indexed to thoracic vertebral bodies starting at the cranial edge of T4 and summed. The VHS was measured to the nearest 0.1 vertebra. 

Based on the VHS, a method to measure the cardiac silhouette indexed to the thoracic inlet is presented. The length of the long and short axes of the cardiac silhouette, measured as described previously for the VHS, were summed and divided by the corresponding thoracic inlet length (TI). The TI is the distance extending from the cranio-ventral aspect of the first thoracic vertebra to the craniodorsal manubrium at its highest point, the point of the minimum length of the thoracic inlet. A unitless value, the Thoracic Inlet Heart Score (TIHS), is obtained (Figure 1). 

Both VHS and TIHS were measured from the right lateral projection on images in DICOM format using a digital caliper in a commercial viewing system (IntechForView 12.5.1.1, La Cartuja Baja, Zaragoza, Spain). Three measurements were made for each method and the average was used for statistical analysis. 

Echocardiography and lateral radiographs were made within 24 h by the first author (DM), a PhD student with over 20 years of clinical experience and echocardiography expertise, who measured the VHS and TIHS values on every patient. This investigator was not blinded to the clinical data and echocardiographic measurements at this time. 

To assess the intra and interobserver variability, two observers (DM and VG, a general practitioner with less than five years of clinical practice) assessed the radiographs of 16 randomly selected dogs independently. The same radiograph was evaluated two times at least one week apart. VG was blinded to the clinical status of each dog and the measurements determined by the other investigator. The interobserver coefficient of variation (CV) was measured by pairing the first measurements of both observers. 

### 2.3. Statistical Analysis

Descriptive variables (age, body weight) were reported as the median and the range (minimum and maximum values). To determine the influence of body weight on the TIHS and VHS, the dogs were divided into four groups (≤10 kg, 10.01–20 kg, 20.01–30 kg, ≥30.01 kg). The variables of interest (TIHS and VHS) were reported as the median, the standard deviation, and the range. A 95% confidence interval (CI) was calculated for the selected measurements, TIHS and VHS. A paired Student’s t test was performed to identify differences in the TIHS values between male and female subjects, between different body weights, and differences in VHS values depending on both sex and body weight. Differences with a *p*-value < 0.01 were considered significant. Pearson’s correlation was performed to assess the relationship between the TIHS and VHS values, *p* < 0.05. In addition, the correlations of the TIHS with the cardiac long axis (LAx), and cardiac short axis (SAx) were assessed. The correlation of the TI length with the fourth thoracic vertebra length (T4) was assessed, as well as the correlation between TI and body weight, TI and LAx, TI and SAx, TI and the sum of LAx and SAx, body weight and LAx, and body weight and SAx. The correlation of VHS and LAx and SAx was also assessed. The correlation was considered weak, moderate, strong, or perfect, when the value of the correlation was 0.1–0.3, 0.4–0.6, 0.7–0.9 or 1, respectively. 

For the intraobserver and interobserver variability, a Kappa agreement (K [95% confidence interval]) was interpreted as: slight agreement (0.01–0.2), fair (0.21–0.40), moderate (0.41–0.60), substantial (0.61–0.80), and almost perfect agreement (0.81–0.99). All statistical analyses were performed using commercially available software (SAS/STAT software, version 16.5, Microsoft Excel 2021). 

## 3. Results

One hundred and forty-four dogs over 1 year of age took part in the study. Breeds represented were 62 mixed-breed, four each of the breeds Chihuahua, French Bulldog, Golden Retriever, Labrador Retriever, and Yorkshire Terrier; three each of the breeds American Pitbull, American Staffordshire, and German Shepherd Dog; two each of the following: Bull Terrier, Canarian hound, Pug, Scottish Terrier, and West Highland White Terrier; and one each of the following: American Bulldog, Beagle, Bichon, Border Collie, Boston Terrier, Boxer, Cavalier King Charles Spaniel, Check Wolf, Chow-Chow, Dachshund, Dalmatian, Garafian Shepherd, Jack Russell Terrier, Lobo Herreño, Malinois, Pekingese, Pomeranian, Pinscher Miniature, Rottweiler, Schnauzer Miniature, and Spanish Water Dog. For the general population statistical study, 22 dogs were excluded, so as not to include more than four dogs of the same breed (9 Yorkshire Terriers, 8 Chihuahuas, 4 Labrador Retrievers, and 1 French Bulldog). Dogs were included in order of entrance to the study, once four dogs of the same breed had been studied the rest were excluded. Eventually, data from 122 dogs (61 males and 61 females), with a median age of 4 years and 2 months (range 1–16 years) and mean body weight of 8.39 kg (1.8–48.5 kg) were statistically analyzed. For the body weights: 52/122 (42.6%) dogs weighed less than 10 kg, 26/122 (21.3%) weighed between 10.1 kg and 20 kg, 28/122 (22.9%) weighed between 20.1 kg and 30 kg, and 16/122 (13.1%) weighed over 30 kg. 

Normally distributed, the TIHS value for the overall population was 2.86 ± 0.27 (Table 1). The TIHS value did not differ depending on sex or body weight, *p*=0.96 and *p* > 0.01, respectively. The VHS value for the overall population was 10.12 ± 0.92. The VHS values did not show significant differences depending on sex, *p* = 0.49. The VHS value was lower for the ≤ 10kg group (9.77 ± 0.68) compared to the general population (10.12 ± 0.92) and the other groups (10.01–20 kg was 10.42 ± 1.20; 20.1–30 kg was 10.32 ± 1.01; and ≥30 kg was 10.60 ± 1.12), *p* < 0.01 (Figure 2). The TIHS value was less than 3.25 in 90% of the dogs. The VHS value was ≤ 10.5v in 72% of the dogs, and ≤ 11.5v in 92% of the dogs.

Only one dog presented vertebral malformations, a male 10-year French bulldog weighing 13.6 kg. Excluding French bulldogs, the TIHS and VHS values were 2.87 ± 0.27 and 10.05 ± 0.83, respectively.

TIHS and VHS values were calculated for the four most represented breeds in our hospital: Yorkshire Terrier (13), Chihuahua (12), Labrador Retriever (8), and French Bulldog (5) (Table 2). The Yorkshire Terrier TIHS value showed no differences with the general population, *p* = 0.10. On the contrary, the VHS value for the Yorkshire Terrier breed dogs was significantly different to that of the general population and Labrador Retrievers, *p* < 0.01. The Chihuahua TIHS and VHS values showed no differences compared to the general population, *p* = 0.30 and *p* = 0.09, respectively. The Labrador Retriever TIHS and VHS values showed no differences compared to the general population, *p* = 0.013 and *p* = 0.23, respectively. The French Bulldog TIHS value was significantly shorter compared to the general population, Yorkshire Terrier, and Labrador Retriever, *p* < 0.01. The French Bulldog VHS value was significantly higher than the VHS for the general population, Yorkshire Terrier, Chihuahua, and Labrador Retriever, *p* < 0.01 (Figure 3).

There was a correlation between TI and T4: *r* > 0.9, *p* < 0.05. There was also a correlation between TI and LAx, SAx, and the sum of LAx and SAx: *r* > 0.9, *p* < 0.05. There was a correlation between T4 and LAx and SAx: *r* > 0.9, *p* < 0.05. There was a strong correlation between TI and body weight: *r* > 0.7, *p* < 0.05; and between T4 and body weight: *r* > 0.7, *p* < 0.05. The correlation between body weight and LAx and SAx was *r* > 0.7 and *p* < 0.05. There was not a correlation between TIHS and LAx and SAx: *r* < 0.3, *p* < 0.05. There was not a correlation between VHS and TI: *r* < 0.3, *p* < 0.05. There was not a correlation between VHS and LAx and SAx: *r* < 0.3, *p* < 0.05. TIHS and VHS showed no correlation: *r* < 0.24, *p* < 0.05. 

Intraobserver variability showed almost perfect agreement for TIHS and VHS values, 0.93 and 0.99, respectively. Interobserver variability showed substantial agreement for this values, 0.77, and almost perfect agreement for VHS values, 0.85 (Figure 4).

## 4. Discussion

This study was carried out to describe a radiographic method to assess the heart size normalized to the thoracic inlet length. It shows that TIHS is a simple, straightforward, reliable, and reproducible way to measure a dog’s cardiac silhouette on a thoracic right lateral radiographic projection. The TI length defined in the present study was the distance extending from the craniodorsal manubrium at its highest point to the cranioventral aspect of the first thoracic vertebrae, resulting in the shortest distance. The same thoracic inlet distance reference points have been used in a recent study comparing the radiographic tracheal diameter at different levels on non-brachycephalic small breed dogs [33].

In 2000, Buchanan [6] found a strong correlation between the sum of the long- and short-axes heart dimensions and a 10 vertebrae reference length in 100 normal dogs of various types, *r* = 0.98, suggesting that those structures were related in a proportional way and considering it a useful reference to evaluate the cardiac silhouette. They also found a good correlation between heart size and the length of three or four sternebrae (r = 0.94 and *r* = 0.95, respectively, *p* < 0.0001), considering it not to be advantageous over vertebral correlations, except in dogs with hemivertebrae or other vertebral abnormalities.

More recently, the manubrium length has been proposed a possible reference value to normalize the measurements of width and height of the cardiac silhouette in small- and large-breed dogs [3]. Although anomalies of the manubrium are uncommon [34], it can adopt different shapes: elongated, bullet-shape, rectangular, and camel head-neck-shape [3]. Dogs with an abnormally shaped manubrium, or where its cranial margin could not be identified, were excluded from that study, and a strong correlation was found between the manubrium length and SAx and LAx [3].

The present study included radiographs of dogs where the manubrium could be clearly identified. It showed a strong correlation between TI and LAx and SAx, and Lax + SAx, *r* = 0.96, 0.96, 0.97, respectively. It also showed a strong correlation between the TI and T4 length, and between TI and body weight, indicating a proportion in body structures that is stable enough not to cause high variations that would make the technique unreliable. 

We observed that 42/122 (34.4%) of dogs studied had a manubrium shape considered as bullet shape or camel head-neck shape, as mentioned before (3). The craniodorsal edge of these manubria seemed to be subjectively wider than the rest of the bone. However, the TIHS value for those dogs was 2.86 ± 0.25 (2.31–3.44), compared to the TIHS value of those with rectangular-shaped manubria: 2.86 ± 0.27 (2.31–3.60), without statistically significant difference, *p* = 0.97. 

Also, some dogs had a sternum with a prominent curvature with respect to the thoracic spine. Those dogs (32/122, 26.2%) had a THIS value of 2.91 ± 0.31 (2.44–3.60). On the other hand, dogs with a regular curvature (90/122, 73.8%) had a THIS value of 2.84 ± 0.25 (2.31–3.39). This difference was not statistically significant, *p* = 0.16. Even though no statistical difference was found with respect to sternum inclination or manubrium shape in our population, further studies might be needed to assess how different sternal angulation with respect to the thoracic spine, or different manubria shapes between breeds or individually, affect the TI length.

This study showed a moderate correlation between the VHS value and the sum of LAx and SAx, *r* = 0.31. On the contrary, a weak correlation and no correlation between VHS and the sum of LAx and SAx in large- and small-breed dogs, respectively, was observed by Mostafa et al. [3]. They suggested that this could be related to the relative variations in the vertebral size and shape among dog breeds, and variations in the intervertebral disk space. In a recent study carried out on pugs, 9/12 did not show typical vertebral body malformation, but irregularly shaped vertebrae, trapezoid vertebral bodies that even caused kyphosis and lordosis in three of them [25]. Our study did not exclude dogs with vertebral malformations; however, only one dog presented thoracic vertebral malformations, a French bulldog with VHS 12.5 v. 

Another potential reason for the weak correlation between VHS and Lax + SAx could be the transformation of the short and long axes dimensions into VHS units, and in the selection of anatomic reference points. 

Correlations between the VHS and MHS values were identified for large-breed dogs but not for small-breed dogs, suggesting that for evaluation of cardiac dimensions in large-breed dogs, VHS may be more convenient, but overall-MHS could be considered for further evaluation in both clinically normal and ill dogs, regardless of breed size [3]. Our study found a weak correlation between the VHS and TIHS values. This could suggest that both methods are not interchangeable. 

The VHS value in the present study, 10.12 ± 0.92 (95% CI, 9.95–10.29), was higher than that found by Buchanan et al. [5], 9.7 ± 0.5, and Greco et al., 9.8 ± 0.6 [35], and in agreement with the results of more recent studies, 10.3 ± 0.8 (95% CI, 10.1–10.5) [3], 10.7 ± 0.65v) [36]. Besides, in the present study, only 72% of dogs had a VHS ≤ 10.5 v compared to 95% in the Buchanan study [5]. These differences could be related to the sample size and the number of dogs of each breed included in the different studies, although the present study and Buchanan did not include more than four dogs of the same breed. Another reason could be the reference point selected for the cardiac long axis, the ventral border of the left main stem bronchus to the most distant ventral contour of the cardiac apex [5] compared to the ventral border of the carina for the long axis in the present study. Also, an adjustable caliper was used by Buchanan et al., compared to a digital caliper in the present study. 

A study by Hannson et al. [27] showed that the VHS method is dependent on an individual selection of reference points, mainly for the long axis; the reference point at the heart base, because of greater complexity and variability of the topographic anatomy compared with other reference points, and difficulty in defining the apex because of superimposition of ribs, skin folds, or cranial parts of the liver, and the transformation of the long and short axes dimensions into VHS units. The VHS is an indicator of heart size in relation to body length expressed as total units of vertebral length to the nearest 0.1 vertebra [5]. The TIHS shares the same reference point but lacks the transformation to vertebral body length. The TIHS method relies on the length of a single reference segment, the TI, which strongly correlated with the cardiac axes, T4 length and body weight. The TIHS could be measured in dogs with mid thoracic vertebral malformations, as it does not need to be transposed onto vertebral bodies. However, malformation in the cranioventral aspect of T1 might render its use unreliable. Also, it is a unitless value that reduces bias, as cardiac axes measures do not need to be transformed into vertebral units. We proposed the thoracic inlet length as an appropriate reference value to normalize the short and long heart axes measured on a thoracic right lateral radiographic projection in dogs.

The present study only measured the TIHS and VHS values on right lateral projections. The original VHS study showed no differences on radiographic projections [5]. Some studies have observed differences between the right and left projections [18,35], suggesting that the increased VHS values in right lateral recumbency may be due to a greater distance of the heart from the radiographic cassette in comparison to left lateral recumbency [35]. As determining the existence of differences between right and left projections was not an objective of this study, we suggest the use of the same projection consistently as recommended by Buchanan and Bucheler [5].

This study only included dogs older than 12 months, in agreement with other studies [8,10,12,14,17,37]. A study on growing dogs of different breeds showed no significant differences in relative heart size at 3, 6, and 12 months, 10.0 ± 0.5, 9.8 ± 0.4, and 9.9 ± 0.6 v [38]. No significant correlation was found between age and vertebral heart scale on 61 healthy Norwich terriers [20]. However, another study suggested that there could be an increase in the VHS value with age, due to the deposition of epicardial fat as dogs age, if this occurs in dogs as it has been shown to in humans [18]. If there is a difference in TIHS with aging, this needs to be investigated. 

Considering body weight, our results did not find differences on TIHS values between different groups (*p* > 0.01). Many other studies also did not find differences on VHS depending on body weight [6,11,12,15,17,18,19,21,22,23,24,39]. However, we found that the VHS value for the ≤ 10 kg group was lower than that of the general population and heavier groups. This is in agreement with Mostafa et al., who found that large dogs (≥16 kg) had a higher VHS (10.7 ± 0.5) than small-breed dogs (≤12 kg) (10.3 ± 0.8), *p* < 0.001 [3]. 

Neither were differences found in the TIHS value between males and females, nor in the VHS value, in agreement with some other studies [5,10,11,12,13,16,19,20,21,22,24,25,39]. However, two different studies have found a higher VHS in females than in males: a multibreed study that showed a female Yorkshire Terrier VHS of 10.2 ± 0.7, versus a value for males of 9.6 ± 0.4, *p* < 0.05 [14]; and a study on Dachshund, where the female VHS was 10.8 versus the male VHS of 9.99, *p* = 0.0002 [18]. In this later study, the authors hypothesized that this occurred because their female population was older, and older animals may deposit more epicardial fat [18]. 

The first VHS study was carried out on 100 dogs with no more than four individuals of the same breed and suggested VHS ≤ 10.5 as a clinically useful limit for normal heart size in most breeds [5]. It also implied that larger numbers of dogs of each breed would be required to determine more precisely normal values for different breeds [5]. With that premise, breed-specific VHS values have been published in the last two decades [7,8,9,10,11,12,13,14,15,16,17,18,19,20,21,22,23,24,25] (Table 3). 

The present study compared VHS and TIHS values for the four most popular breeds in our hospital: Yorkshire Terrier, Chihuahua, Labrador Retriever and French Bulldog. We found differences in the TIHS values between French Bulldogs and the general population, Yorkshire Terriers, and Labrador Retrievers, *p* < 0.01. French Bulldogs have a higher VHS compared to other breeds (14), as has been shown in this study. Considering these results, a larger sample of dogs for each breed would be necessary to ascertain whether different breeds have different this values. 

Intraobserver variability showed almost perfect agreement for both TIHS and VHS, 0.93 and 0.99, respectively, which is in line with previous VHS studies [25,36]. Interobserver variability showed substantial agreement for TIHS, 0.77, and almost perfect agreement for VHS, 0.85. A difference of almost 1 vertebral unit has been observed between observers measuring VHS [27]. However, a more recent study measuring VHS, VLAS, and RLAD in pugs showed that the VHS was the score with the fewest interobserver differences: merely differing between 0.06 to 0.18 v [25], which agrees with several studies showing an interobserver agreement between 0.89 and 0.99 [12,18,20,25,36,40].

A better interobserver agreement for VHS values in our study could be explained because VHS is a well-established and reproducible radiographic score [25], and is commonly used [8,27]. In a study that investigated the reproducibility of several radiographic measurements, and the ease of determining the exact position of their radiographic reference points, the VHS value was independent of the observers’ experience, the cranial contour of the cardiac silhouette being the easiest landmark to locate, with the hardest being the cardiac apex [36]. TIHS is a method that requires the measurement of TI, a segment not usually measured, and it might take some time to identify the reference points. 

This study has some limitations. The main observer was not blinded to the dogs’ clinical status and echocardiography values, which could have caused a radiographic measurement bias. Radiographs where the manubrium was clearly visible were included in the study, although in some patients the manubrium dorsal border was blurry and that could hamper the caliper positioning. The same is true for the cranioventral aspect of the first thoracic vertebrae; in some animals, the superimposition of the first rib with the cranioventral aspect of the vertebra could have misled the authors in the identification of the reference point. 

As mentioned, there was a small interobserver variability while measuring TIHS. We think that this could be related to measuring the TI rather than the LAx and SAx axes, as these two measures were the same for VHS and, the interobserver variability for VHS was less than that for TIHS. As happened with measuring the LAx and SAx to determine the VHS, practitioners need to become familiar with measuring the TI length to calculate TIHS. A study comparing the results of more observers would be desirable.

Although radiographs were intended to be taken during peak inspiration, this is not always possible in awake animals. Moreover, whether there is a change in the thoracic inlet length depending on the respiratory cycle has not been studied, nor how this could affect the TIHS value. 

Though the study included more than 120 animals, as preferable for creating valid reference values in veterinary medicine according to recent guidelines [41], and the number of dogs included for measurements in the present study is higher than those of others studies investigating ranges for VHS [5,11,12,19,20,22], the TIHS value showed here could differ with a larger study. Also, studies on selected breeds might show breed-specific TIHS values, as happens with VHS.

The THIS value results from this study come from a dog population considered to have a normal heart based on history, clinical signs, examination findings, radiography, and echocardiography. Some heart diseases such as aortic stenosis, pulmonic stenosis, mild atrial or ventricular septal defects, early myxomatous mitral valve disease, arrythmias, and endocarditis, do not necessarily change the cardiac size. Studies assessing the usefulness of the TIHS method to discriminate between dogs with and without heart disease are warranted.

## 5. Conclusions

The TIHS method is a feasible, simple, and reliable method to measure the radiographic cardiac silhouette in dogs. With a mean value of 2.86 and a confidence interval of 2.81–2.91 (95%), a TIHS value ≤ 3.2 is suggested as a clinically useful upper limit for the normal heart size for a healthy dog in a general population.

## Figures and Tables

**Figure 1 animals-13-00389-f001:**
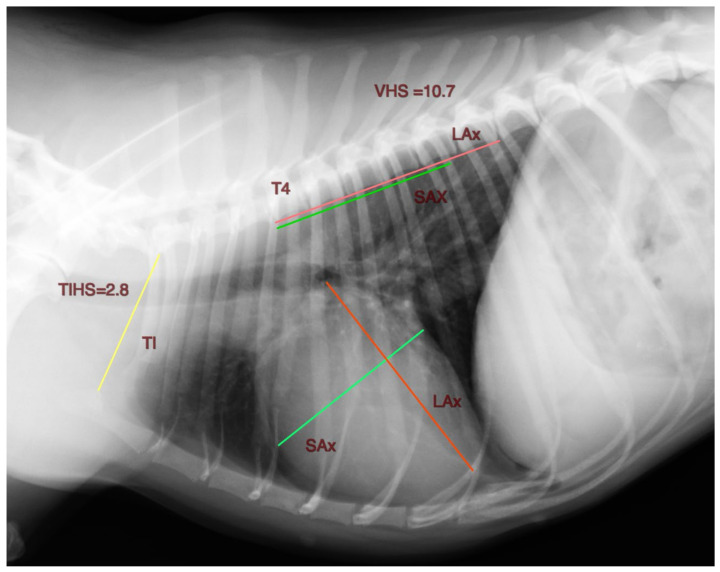
Right lateral thoracic radiographic projection of a clinically normal 9 years old chihuahua illustrating the thoracic inlet heart size measurement method (THIS). The long axis (LAx) and short axis (SAx) of the heart and the thoracic inlet (TI) are measured. The sum of the LAx and SAX is divided by the TI to obtain the thoracic inlet heart size. VHS Vertebral Heart Size.

**Figure 2 animals-13-00389-f002:**
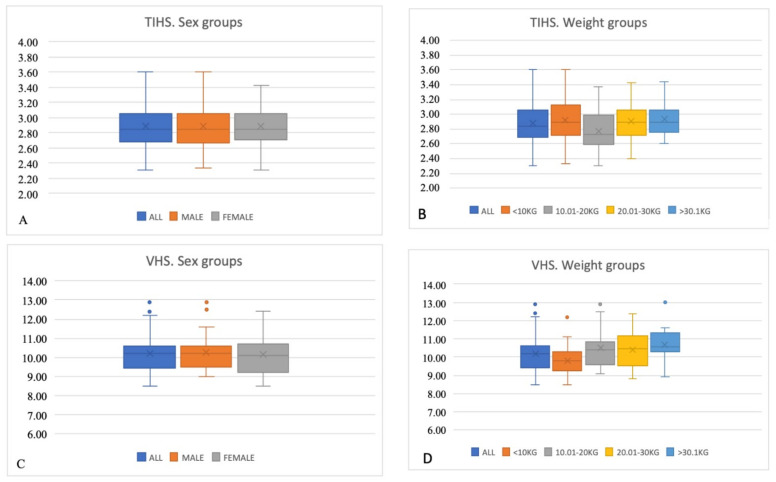
Box plots illustrating TIHS (**A**,**B**) and VHS (**C**,**D**) for the general population, sex groups (male and female), and body weight groups (≤10 kg, 10.01–20 kg, 20.01–30 kg, and ≥30.01 kg). Dots represent outliers.

**Figure 3 animals-13-00389-f003:**
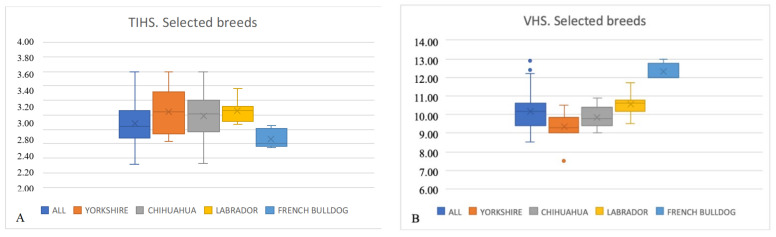
Box plots illustrating TIHS (**A**) and VHS (**B**) for the general population and four most represented dog breeds. Dots represent outliers.

**Figure 4 animals-13-00389-f004:**
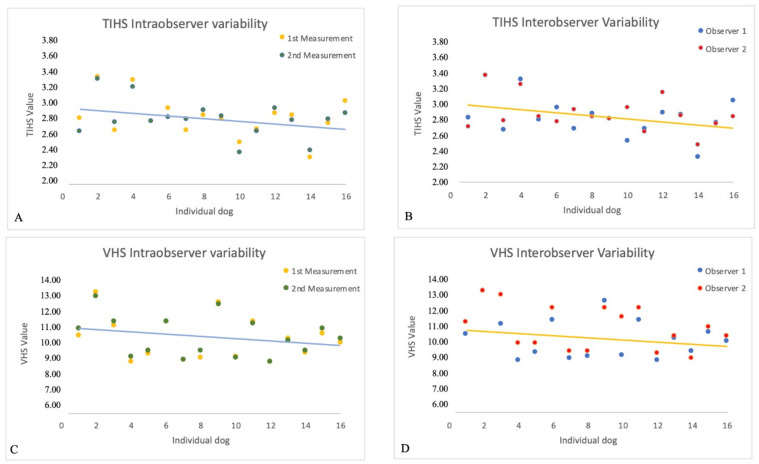
Scatterplots for intra- and interobserver variability on TIHS (**A**,**B**) and VHS (**C**,**D**) values measured on right lateral thoracic radiographs from 16 randomly selected dogs. The solid line represents the tendency line for the variable measured.

**Table 1 animals-13-00389-t001:** Mean (SD), range, and 95% Confidence Interval (CI) for measurements of the cardiac size on a right lateral thoracic radiographic projection obtained from 122 normal dogs. *p* < 0.01. n number of dogs. TIHS Thoracic Inlet Heart Size. VHS Vertebral Heart Size. * Difference statistically significant in VHS value between ≤10 kg group the general population, and the other groups.

		TIHS				VHS			
	*n* ( )	(Mean ± SD)	Range	CI 95%	*p* Value	(Mean ± SD)	Range	CI 95%	*p* Value
General population	122	2.86 ± 0.27	2.31–3.60	2.81–2.91		10.12 ± 0.92	8.5–13.0	9.95–10.29	
Sex	Male (61)	2.86 ± 0.26	2.33–3.60	2.77–2.95		10.18 ± 0.92	9.0–13.0	9.88–10.48	
	Female (61)	2.85 ± 0.27	2.31–3.43	2.76–2.94	0.96	10.05 ± 0.95	8.5–12.40	9.74–10.36	0.49
Weight	≤10 kg (52)	2.89 ± 0.27	2.33–3.60	2.77–2.99	0.41 (GP) 0.03 (10.01–20) 0.77 (20.01–30) 0.77 (>30)	9.77 ± 0.68	8.5–12.20	9.66–10.22	0.007 (GP) * 0.0011 (10.01–20) * 0.0017 (20.01–30) * 0.0001 (>30) *
	10.01–20 kg (26)	2.75 ± 0.27	2.31–3.37	2.60–2.90	0.48 (GP) 0.07 (20.01–30) 0.07 (>30)	10.42 ± 1.20	9.1–13.0	9.81–11.03	0.31 (GP) 0.71 (20.01–30) 0.64 (>30)
	20.01–30 kg (28)	2.88 ± 0.24	2.40–3.43	2.76–3.00	0.73 (GP) 0.59 (>30)	10.32 ± 1.01	8.80–12.40	9.83–10.81	0.05 (GP) 0.33 (>30)
	≥30.1 kg (16)	2.92 ± 0.24	2.61–3.44	2.76–3.10	0.40 (GP)	10.60 ± 1.12	8.90–13.00	9.88–11.32	0.05 (GP)

**Table 2 animals-13-00389-t002:** Mean (SD), range, and 95% Confidence Interval (CI) for measurements of the cardiac size on right lateral thoracic radiographic projection obtained from 48 normal dogs of four different breeds. *p* < 0.01.

		TIHS				VHS			
Breeed	*n*	(Mean ± SD)	Range	CI 95%	*p* Value	(Mean ± SD)	Range	CI 95%	*p* Value
Yorkshire terrier	13	3.01 ± 0.31	2.64–3.34	2.79–3.23	0.10 (GP) 0.70 (Chihuahua) 0.73 (Labrador Retriever)	9.28 ± 0.74	9.0–10.5	8.75–9.81	0.002 (GP) ** 0.07 (Chihuahua) 0.003 (Labrador Retriever) **
Chihuahua	12	2.96 ± 0.33	2.73–3.60	2.71–3.19	0.30 (GP) 0.45 (Labrador Retriever) 0.011 (French Bulldog)	9.82 ± 0.60	9.10–10.90	9.38–10.26	0.09
Labrador retriever	8	3.05 ± 0.16	2.87–3.36	2.91–3.19	0.013 (GP)	10.54 ± 0.63	10.57–11.94	9.97–11.11	0.23
French Bulldog	5	2.66 ± 0.13	2.55–2.86	2.41–2.81	0.0028 (GP) * 0.0027 (Yorkshire Terrier) * 0.0007 (Labrador Retriever) *	12.29 ± 0.45	12.00–13.00	11.77–12.81	0.0012 (GP) ** 0.0000 (Yorkshire Terrier) ** 0.0000 (Chihuahua) ** 0.0002 (Labrador Retriever) **

GP: General Population. n: number of dogs. THIS: Thoracic Inlet Heart Size. VHS: Vertebral Heart Size. * Difference statistically significant in TIHS value between French Bulldog breed dogs and the general population, Yorkshire Terrier and Labrador Retriever. ** Difference statistically significant in VHS value between Yorkshire Terrier and the general population, and Labrador Retriever; and in VHS value between French Bulldogs and the general population, Yorkshire Terrier, Chihuahua and Labrador Retriever dogs.

**Table 3 animals-13-00389-t003:** Breed-specific VHS obtained in different studies since the original Vertebral Heart Size study was published by Buchanan and Bücheler in 1995.

Breed	Number	VHS	Study
American Pitbull Terrier	24	10.9 ± 0.4 (10.5–11.8)	[26]
Australian cattle dog	20	10.5 ± 0.5 (9.8–11.3)	[19]
Beagle	19	10.3 ± 0.4 (9.2–11.2)	[13]
Belgian Malinois	19	9.58 ± 0.53 (8.52–10.35)	[16]
Boston Terrier	19	11.4 ± 1.2 14.2 ± 1.6 (Vertebral anomalies)	[14]
Boxer	22	11.3 ± 0.8 (10.3–12.6)	[8]
Brittany Spaniel	28	10.6 ± 0.2	[23]
Bulldog	30	12.1±1.5 13.4 ± 1.6 (Vertebral anomalies)	[14]
Cavalier King Charles Spaniel	20	10.6 ± 0.5 (9.9–11.7)	[8]
Chihuahua	30	10.0 ± 0.6 (8.9–11.0)	[21]
Dachsund	29 51	9.7 ± 0.5 10.3 (9.25–11.55)	[14]
Doberman	20	10.0 ± 0.6 (9–10.8)	[8]
German Shepherd Dog	21 100	9.9 ± 0.7 (8.7–11.2) 9.8 ± 0.5 (9.2–10.3)	[8] [7]
Greyhound	42	10.5 ± 0.1	[12]
Indian Spitz	20	10.21 ± 0.23	[17]
Labrador	19 24 20	10.8 ± 0.6 (9.7–11.7) 10.39 ± 0.5 10.22 ± 0.20	[8] [15] [17]
Lhasa apso	18	9.6 ± 0.8	[14]
Maltese	81	9.53 ± 0.4	[22]
Norwich Terrier	61	10.6 ± 0.60	[20]
Poodle	30	10.12 ± 0.51 (9.2–11.1)	[9]
Pomeranian	18	10.5 ± 0.9	[14]
Pug	30 32	10.7 ± 0.9 11.25 ± 0.62 (10.1–12.8)	[14] [25]
Rottweiler	38	9.8 ± 0.1	[12]
Shih-Tzu	30	9.5 ± 0.6	[14]
Turkish Shepherd Dog	120	9.7 ± 0.67 (8.4–10.9)	[11]
Whippet	44	11.03 ± 0.5 (10.1–11.8)	[10]
Yorkshire Terrier	12 30	9.7 ± 0.5 (9.0–10.5) 9.98 ± 0.6	[8] [14]

## Data Availability

The data presented in this study are available on request from the corresponding author.

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
