# Peer review of "The Thoracic Inlet Heart Size, a New Approach to Radiographic Cardiac Measurement"

_animals, 2023, doi:10.3390/ani13030389_

Round 1

Reviewer 1 Report

Thank you for letting me review this interesting paper focused  on the hypothesis that the thoracic inlet is a suitable reference point for the assessment of heart size on right lateral radiographic projection in dogs. 

First of all the paper is coherent to the aim and scope of the journal.

I have just a minor change to suggest to the authors. Materials and Method are well described, but if possible they should add supplementary information about the weight not just > 15kg or < 15 kg maybe it better to use , length of the animal so the study will be more reliable.

Author Response

Thank you for reviewing our paper. We have read your suggestions and have made a change to the paper that we hope helps to make the study more reliable. 

You commented: I have just a minor change to suggest to the authors. Materials and Method are well described, but if possible they should add supplementary information about the weight not just > 15kg or < 15 kg maybe it better to use , length of the animal so the study will be more reliable.

-We agree that groups of less than 15kg and over 15kg can be very heterogenous. We have changed that part of the statistical analysis. We divided the dogs in four groups (≤10kg, 10.01-20kg, 20.01-30kg, ≥30.01kg. Line 167; 201-203. Table 1. Figure 2.

-Making those four body weight groups we found differences on the VHS between the ≤10kg group and the general population and with the rest of the groups. Lines 348-349.

We hope that change adds value to the content of our paper and makes it more clear to readers. 

Than you.

Reviewer 2 Report

Based on previously and standardized measurements (LAx, SAx and TI), the authors developed a new calculation (TIHS=LAx+SAx/TI), with the aim of evaluating the heart size radiographically in dogs.

The authors were careful in dog recruitment and taking thoracic radiographic views. In addition, they evaluated a considerable number of dogs (N=122). The quality of the thoracic radiographic view is fundamental for the determination of heart size by both VHS and TIHS. However, vertebral malformations and conversion to vertebral units seem to be the main disadvantages of VHS compared to TIHS.

The results suggest that TIHS can be used as a way to assess the heart size radiographically, but some points should be considered by the authors:

Introduction (lines 41-42): "However, the skill and expertise to perform and interpret echocardiographic examinations is often not available [2]."The same statement is valid regarding the ability and experience to measure the long and short cardiac axis and the length of the thoracic inlet in the radiographic views.

Introduction (lines 85-86): The authors mentioned that "The objective of this prospective study was to establish a new method to assess the heart size radiographically...". Is this a new method? Or an adaptation/modification of previously developed method(s)?

It is noteworthy that the researchers established the TIHS value from measurements and standards described by other authors, mainly Buchanan and Bücheler (1995).

The reason for measuring TIHS and VHS in the right lateral view was mentioned in the Discussion (lines 319-326) but could have been indicated in Material and Methods.

Dogs with a body weight of around 15kg can be considered as medium dogs. Did the authors compare other body weight ranges (eg < 10kg, 10.1 - 20kg, 20.1 - 30kg, > 30kg)?

In Pearson's correlation, was the p value < 0.01?

A review of the correlation results is recommended:

In the results, the correlations are shown, but without the p-values (lines 228-236). Statistically, the relevant correlations are those that are closer to -1 or 1 and with a p-value <0.05. Better than presenting weak, moderate or strong correlation results, it would be to indicate whether or not there was a correlation between the variables (r>0.7 and p-value <0.05). The TI, T4, LAx and SAx variables were determined in mm or cm and could be correlated with each other. On the other hand, it is difficult to expect a correlation between vertebral units (VHS) and variables in mm or cm (TI, T4, LAx and SAx).

Discussion (lines 313-315): "TIHS could be measured in dogs with and without vertebral malformations, as it does not need to be transposed onto vertebral bodies." Is it a fact that dogs with vertebral malformations do not show alterations in the cranio-ventral aspect of the first thoracic vertebra and consequently in the TI distance?

The authors did not assess the influence of BCS on TIHS; however, they discuss VHS values in dogs with different BCS (lines 341-346).

No relevant correlation was observed between VHS and TIHS, therefore, the authors suggested that VHS and TIHS are not interchangeable (lines 291-292). Even so, they argued a little too much about VHS results (lines 349-368).

Conclusion (lines 412-413): "TIHS is a simple, reliable and reproducible method to measure the radiographic cardiac silhouette in dogs." Considering that there was a small interobserver variability for the TIHS, this statement may be valid mainly for professionals with experience in measuring LAx, SAx and TI.

Author Response

Thank you for reviewing our paper. We have read your suggestions and have made some changes to the paper that we hope helps to make the study more reliable. 

You commented:

Introduction (lines 41-42): "However, the skill and expertise to perform and interpret echocardiographic examinations is often not available [2]."The same statement is valid regarding the ability and experience to measure the long and short cardiac axis and the length of the thoracic inlet in the radiographic views.

In our opinion, the training required to perform an echocardiography is much more that the training needed to take three measures, in our method, on an x-ray. We consider more time consuming transforming a length to vertebral units than to do a division. The idea behind the TIHS method is to provide the general practitioner with another tool to evaluate the cardiac silhouette on radiographies, simpler to perform from our point of view than VHS. We have found some references on the literature that go in the direction of providing the first opinion veterinarian with simple tools to recognize cardiac enlargement on x-rays:

-JW Buchanan in Vertebral scale system to measure heart size in radiographs, Clinical radiology Veterinary Clinics of North America: Small Animal Practice Vol. 30. Nº2, March 2000, states that “cardiac mensuration is helpful for inexperienced observers as a starting point in evaluating heart size”.

-Also, this more recent study Use of cardiac sphericity index and manubrium heart scores to assess radiographic cardiac silhouettes in large-
and small-breed dogs with and without cardiac disease
AA. Mostafa, K E.. Peper, Clifford R. Berry. Journal of the American Veterinary Association,  Vol. 256. Nº. 8 888-898. Apr. 2020 considers that “Although trained personnel can often recognize cardiomegaly evident on thoracic radiographic images, inexperienced veterinarians may benefit from a quantitative procedure to assess the size and shape of the cardiac silhouette”.

Nevertheless, we have eliminated that statement as some training is also required to assess the heart size in radiographies.

Introduction (lines 85-86): The authors mentioned that "The objective of this prospective study was to establish a new method to assess the heart size radiographically...". Is this a new method? Or an adaptation/modification of previously developed method(s)?

We agree that this is an adaptation of a previously developed method. In order not to create controversy we deleted the word “new” in line 85. Also, in Materials and Methods Radiography line 137 we commenced the description of the TIHS method by saying “Based on the VHS method”. However, the word “new has been deleted on line 137, 261 and 404.

It is noteworthy that the researchers established the TIHS value from measurements and standards described by other authors, mainly Buchanan and Bücheler (1995).

Based on Buchanan VHS other method have been described and introduced to the reader as new.

A new method of computing the vertebral heart scale by means
of direct standardization.
X. Sánchez, D. Prandi, L. Badiella, A. Vázquez, F. Llabrés-Díaz, C. Bussadori, O. Domènech. Journal of Small Animal Practice (2012) 53, 641–645 doi: 10.1111/j.1748-5827.2012.01288.x. 

The aim of this study was to describe and compare an adapted and simplified VHS method (the objective VHS) with the original Buchanan VHS method used in previous publications. Providing clinicians with a precise description of how to accurately measure L, S and T4-T8, as well as with the direct standardisation formula, would decrease the variability affecting the quantification of the VHS value and would offer reliable results.

The heart to single vertebra ratio: A new objective method for radiographic assessment of cardiac silhouette size in dogs D. Costanza, A. Greco, D. Piantedos1, D. Bruzzese, M.P. Pasolini, P. Coluccia, E. Castiello, C. S. Baptista, L. Meomartino. Vet Radiol Ultrasound. 2022;1–7.

We hypothesized that a single vertebra preserves its proportion to the respect of the whole body as well as the thoracic vertebral tract proposed by Buchanan and Bucheler.8 The use of a single vertebra, without shape and dimensions alterations, could allow the clinician to objectively evaluate cardiac silhouette dimensions, even in patients with thoracic spine alterations. Therefore, the primary aim of this study was to develop a novel method, termed the heart-to-single vertebra ratio (HSVR). Secondary objectives were to test the level of agreement between the newly described method and VHS, and to evaluate the intra- and inter-observer agreement among three observers with different levels of experience. Then, the cardiac long axis (LA) and short axis (SA) were measured as described by Buchanan and Bücheler.

The reason for measuring TIHS and VHS in the right lateral view was mentioned in the Discussion (lines 319-326) but could have been indicated in Material and Methods.

We added in Material and Methods that a right lateral projection is the projection that we usually take in our hospital. Line 118-119.

Dogs with a body weight of around 15kg can be considered as medium dogs. Did the authors compare other body weight ranges (eg < 10kg, 10.1 - 20kg, 20.1 - 30kg, > 30kg)?

We agree that groups of less than 15kg and over 15kg can be very heterogenous. We did compare other weights, < 10kg, 10.1 - 20kg, 20.1 - 30kg, > 30kg, and found no differences. We have changed that part of the statistical analysis. We divided the dogs in four groups (≤10kg, 10.01-20kg, 20.01-30kg, ≥30.01kg. Line 167. Table 1. Figure 2.

In Pearson's correlation, was the p value < 0.01?

In Pearson’s correlation p value was p<0.05. Added to Statistical analysis, line 174.

A review of the correlation results is recommended:

In the results, the correlations are shown, but without the p-values (lines 228-236). Statistically, the relevant correlations are those that are closer to -1 or 1 and with a p-value <0.05. Better than presenting weak, moderate or strong correlation results, it would be to indicate whether or not there was a correlation between the variables (r>0.7 and p-value <0.05). The TI, T4, LAx and SAx variables were determined in mm or cm and could be correlated with each other. On the other hand, it is difficult to expect a correlation between vertebral units (VHS) and variables in mm or cm (TI, T4, LAx and SAx).

We have considered the suggestion made and instead of using words like strong, moderate, weak when talking about correlation we added the r values. Lines 238-245.

Considering the different units when comparing measures we observed correlation between body weight (expressed in kg) and LAx and SAx and TI (express in mm), r>0.7

Discussion (lines 313-315): "TIHS could be measured in dogs with and without vertebral malformations, as it does not need to be transposed onto vertebral bodies." Is it a fact that dogs with vertebral malformations do not show alterations in the cranio-ventral aspect of the first thoracic vertebra and consequently in the TI distance?

We agree. Amended, lines 337-339.

The authors did not assess the influence of BCS on TIHS; however, they discuss VHS values in dogs with different BCS (lines 341-346).

We agree with that so we have eliminated that part from the discussion.

No relevant correlation was observed between VHS and TIHS, therefore, the authors suggested that VHS and TIHS are not interchangeable (lines 291-292). Even so, they argued a little too much about VHS results (lines 349-368).

This is to show that VHS is influenced by many factors, and this might happen with TIHS, it needs further study. However, we have deleted some information as suggested, as the study is about THIS and not VHS results. Lines 382.

Conclusion (lines 412-413): "TIHS is a simple, reliable and reproducible method to measure the radiographic cardiac silhouette in dogs." Considering that there was a small interobserver variability for the TIHS, this statement may be valid mainly for professionals with experience in measuring LAx, SAx and TI.

It is hard to say how much time it is required to gain enough experience to measure LAx, SAx and TI. Determining the reference point measurement learning curve would require a dedicated study. Although, one of the observers of this study had less than five years of experience and found no difficulties in measuring LAx. SAx and TI. We agree that there is a small interobserver variability while measuring TIHS. We think that this could be related to measuring the TI rather than the LAx and SAx axes, because these were the same for VHS and the interobserver variability for VHS was less than that for TIHS. Practitioners need to get familiar with measuring the TI length. Further studies might be needed to confirm that the method is reproducible by many observers. Lines 414-419.

We addressed the aforementioned sentence by removing the word reproducible. Line 437.

We hope these changes add value to the content of our paper and makes it more clear to readers. 

Thank you.

Reviewer 3 Report

Thank you for providing the opportunity to review your submission. The study describes a new evaluation method (thoracic inlet heart size: TIHS) for heart size in dogs. I believe that all conclusions can be supported by available data. I have outlined the supporting data below. The authors may be able to salvage information of publishable quality after considerable reworking of the available data.

Major comments

This study has one major problem. As the authors described in manuscript, in some breeds, vertebral heart size (VHS) assessment is challenging. In this study TIHS was calculated in three breeds, Yorkshire Terrier, Chihuahua, and Labrador Retriever. Why was the statistical analysis performed between each breed and the general population value? Should it be performed among the breeds? Moreover, VHS is overestimated in brachycephalic dogs due to thoracic or vertebral malformations. To fully utilize TIHS, it should be calculated in brachycephalic breeds.

Minor comments

Line 236: It was described that the correlation between TIHS and VHS was weak. I am asking out of curiosity. If TIHS is within the reference range and VHS exceeds the upper limit, or if TIHS exceeds the upper limit and VHS is within the reference range, how will you judge that?

Lines 167, 168, and 294: extra space

Line 253: no period 

Author Response

We thank you for reviewing our pape. We have read with keen interest your comments and suggestions and made some changes to the paper that could add value to it and make it clearer to the reader. 

Your comments:

1) Major comments

- This study has one major problem. As the authors described in manuscript, in some breeds, vertebral heart size (VHS) assessment is challenging. In this study TIHS was calculated in three breeds, Yorkshire Terrier, Chihuahua, and Labrador Retriever. Why was the statistical analysis performed between each breed and the general population value? Should it be performed among the breeds? Moreover, VHS is overestimated in brachycephalic dogs due to thoracic or vertebral malformations. To fully utilize TIHS, it should be calculated in brachycephalic breeds.

We have added the statistical analysis among the four most popular breeds in our hospital. Lines 223. Table 2. Figure 3.

We only had eight brachycephalic dogs in our study. Five French Bulldogs, 2 Pugs and 1 Boxer. We compared the TIHS value of French Bulldogs with the TIHS value of the general population and the other three breeds. We found that there was a statistically significant difference in the THIS value between French Bulldogs and the other breeds, and with the general population.

Lines 230-233. Table 2. Figure 3.

We consider that breed-specific TIHS studies would be needed to assess breed variability. Lines  387-389.

2) Minor comments

- Line 236: It was described that the correlation between TIHS and VHS was weak. I am asking out of curiosity. If TIHS is within the reference range and VHS exceeds the upper limit, or if TIHS exceeds the upper limit and VHS is within the reference range, how will you judge that?

This is a very good question. In our population of 122 healthy dogs without heart disease, 41/122 (33%) dogs had a VHS ≥10.5 considered the upper limit by Buchanan and Bücheler in 1995. Of those 41 dogs, 9 dogs (21,9%) had a THIS >3.2, being 3.2 the TIHS upper reference value from our results. On the other hand, 15/122 (12,3%) dogs had a THIS value >3.2, and 11/15 (73,3%) dogs with THIS>3.2 had a VHS≥10.5. From these results we would measure both and considered TIHS as more reliable. Although, as it happens in veterinary medicine, a diagnostic tool adds valuable information to the clinical case, but its results should not be considered alone but together with the information gathered through the history, clinical signs, examination findings and other diagnostic methods. If VHS exceeds the upper limit for the general population or the breed and TIHS is normal, or vice versa I would take a look to the general picture; does the patient belong to a dog breed predisposed to heart disease, considering its age, is it a puppy with a continuous heart murmur or a middle age toy breed dog with an apical heart murmur, or a large dog with syncope, arrythmia, then I would recommend further diagnosis, for instance an echocardiography.

- Lines 167, 168, and 294: extra space

Corrected.

- Line 253: no period 

Corrected.

Thank you very much.

Round 2

Reviewer 3 Report

No comment.